# Outcomes of Patients Treated with Blood Transfusion in a Contemporary Tertiary Care Medical Center Intensive Cardiac Care Unit

**DOI:** 10.3390/jcm12041304

**Published:** 2023-02-07

**Authors:** Hani Karameh, Sharon Bruoha, Louay Taha, Meir Tabi, Rivka Farkash, Mohammad Karmi, Kamal Hamayel, Nimrod Perel, Yoed Steinmetz, David Marmor, Mohammed Manassra, Tomer Maller, Rafael Hitter, Itshak Amsalem, Michael Glikson, Elad Asher

**Affiliations:** 1Jesselson Integrated Heart Center, Shaare Zedek Medical Center and Faculty of Medicine, Hebrew University of Jerusalem, Jerusalem 9103102, Israel; 2Department of Cardiology, Barzilai Medical Center, Ben-Gurion University of the Negev, Beer Sheva 84105, Israel

**Keywords:** blood transfusion, ICCU, mortality

## Abstract

Background: Acutely ill patients treated with blood transfusion (BT) have unfavorable prognoses. Nevertheless, data regarding outcomes in patients treated with BT admitted into a contemporary tertiary care medical center intensive cardiac care unit (ICCU) are limited. The current study aimed to assess the mortality rate and outcomes of patients treated with BT in a modern ICCU. Methods: Prospective single center study where we evaluated mortality, in the short and long term, of patients treated with BT between the period of January 2020 and December 2021 in an ICCU. Outcomes: A total of 2132 consecutive patients were admitted to the ICCU during the study period and were followed-up for up to 2 years. In total, 108 (5%) patients were treated with BT (BT-group) during their admission, with 305 packed cell units. The mean age was 73.8 ± 14 years in the BT-group vs. 66.6 ± 16 years in the non-BT (NBT) group, *p* < 0.0001. Females were more likely to receive BT as compared with males (48.1% vs. 29.5%, respectively, *p* < 0.0001). The crude mortality rate was 29.6% in the BT-group and 9.2% in the NBT-group, *p* < 0.0001. Multivariate Cox analysis found that even one unit of BT was independently associated with more than two-fold the mortality rate [HR = 2.19 95% CI (1.47–3.62)] as compared with the NBT-group, *p* < 0.0001]. Receiver operating characteristic (ROC) curve was plotted for multivariable analysis and showed area under curve (AUC) of 0.8 [95% CI (0.760–0.852)]. Conclusions: BT continues to be a potent and independent predictor for both short- and long-term mortality even in a contemporary ICCU, despite the advanced technology, equipment and delivery of care. Further considerations for refining the strategy of BT administration in ICCU patients and guidelines for different subsets of high-risk patients may be warranted.

## 1. Introduction

With advanced interventions and delivery of care, modern intensive cardiac care units (ICCUs) have changed from units primarily dedicated to managing patients with acute coronary syndromes (ACS) into units dealing with heterogeneous patients with different acute and chronic cardiovascular conditions [1,2]. Consequently, the prevalence of ischemic heart disease patients has decreased, whereas the proportion of patients with other cardiac comorbidity conditions such as valvular heart disease, heart failure (HF), malignant arrythmias and venous thromboembolism [3] increased. Moreover, not uncommonly, patients with chronic cardiovascular comorbidities presenting with acute noncardiac conditions, such as sepsis and kidney injury, are also managed in the ICCU [2,4].

Anemia commonly affects critically ill patients and is associated with increased morbidity and mortality [5]. The management of low hemoglobin levels frequently includes blood transfusions (BT), one of the most common treatments in the intensive care setting [6]. However, the benefits of liberal BT in hemodynamically stable trauma patients and critically ill patients with anemia are debatable [7,8,9,10,11]. Conversely, patients with pre-existing coronary lesions and limited flow reserve may experience a higher rate of adverse events due to the additional risk of myocardial ischemia and infarction as a result of anemia [12]. In the recently published REALITY trial, evaluating patients with acute myocardial infarction (MI) and anemia, a restrictive compared with a liberal transfusion strategy resulted in a noninferior rate of major adverse cardiac events (MACE) after 30 days [13]. Despite being a potentially lifesaving therapy, BT may be associated with fatal adverse events including transfusion-associated circulatory overload (TACO) and transfusion-related acute lung injury, which are associated with non-neglectable mortality, apart from other infectious and non-infectious transfusion reactions including transfusion-associated immunosuppression and allosensitization, which is known as transfusion-related immune modulation (TRIM) [12,14,15,16,17,18]. Despite several clinical practice guidelines for transfusion, none fully addresses the relevant subgroups of ICCU patients [18,19,20]. Available evidence supports a restrictive BT approach for non-bleeding, critically ill, general ICU patients with hemoglobin (Hb) level <7 g/dL. In patients with coexisting cardiovascular disease and planned cardiac surgery, the cut-off is higher at Hb level of <8 g/dL [15,18,21,22,23,24,25]. Hence, the aim of the current study was to examine mortality rates and outcomes in patients treated with BT in a contemporary tertiary ICCU.

## 2. Methods

All patients admitted to the ICCU at a tertiary care medical center between January 2020 and December 2021 were included in the study and were followed-up for up to 2 years. Restrictive transfusion strategy was adopted at out center, where BT was indicated if Hb level was ≤7–8 g/dL or if there was evidence of active bleeding with signs of hemodynamic instability [13,26,27]. Hemodynamic instability was defined as persisting hypotension requiring vasopressors to maintain mean arterial pressure ≥65 and serum lactate level greater than 2 mmol/L in the absence of hypovolemia at the time of transfusion [28].

### 2.1. Data Collection

Data were anonymously submitted and prospectively documented into an electronic case report form and then were checked for accurateness and out-of-range values by a local coordinator. Demographic data, comorbidity conditions and clinical diagnoses were systematically recorded as well as laboratory and imaging data. Overall mortality rates were determined from the Israeli Ministry of Internal Affairs data.

The study was approved by the SZMC Institutional Review Board (approval number 0233-19-SZMC) with the exemption from informed consent.

### 2.2. Study Outcomes

The primary outcome was overall mortality rates, assessed at 1 year from time of index hospitalization. The study secondary outcomes were in-hospital complications during the index hospitalization and length of stay.

### 2.3. Statistical Analyses

For categorical variables, characteristics were expressed as numbers and percentages, and for continuous variables, as means, standard deviations or medians with interquartile ranges. Chi-square and Fisher’s exact tests were used to analyze relationships between categorical variables. Through the use of Student’s t- and Mann–Whitney tests, the impact of categorical variables on continuous measurements was examined. The distribution of the continuous variable determined whether a parametric or nonparametric test should be used. By using a stepwise backward Cox proportional hazards model adjusted for blood transfusion groups for age, sex, prior cardiac intervention and different cardiac risk factors, the mortality rate was examined. All tests were two-sided, and statistical significance was set at *p* < 0.05. Version 25.0 of SPSS Statistics for Windows was used to conduct the analyses (IBM Corp, Armonk, NY, USA).

## 3. Results

### 3.1. Patient Characteristics

Two thousand one hundred and thirty-two patients were admitted to the ICCU during the study period. The mean age was 66.9  ±  16 years old and 650 (30.5%) were female. One hundred and eight (5%) patients were treated with BT (BT-group) during their hospitalization, with a total of 305 packed cells (PCs) transfusions. Of them, 31.4% of patients received a single PC, 29.6% received two PCs and 39% received ≥ three PCs, as shown in Figure 1. Patients who received BT were more likely to be elderly, female, have lower body mass index (BMI) and suffer more from hypertension, coronary artery disease (CAD), congestive heart failure (CHF), chronic obstructive pulmonary disease (COPD), pulmonary hypertension, anemia and chronic kidney disease (CKD), compared with the non-BT group. Of note, ninety-five percent of PC transfusions were of standard unmodified packed red blood cells, whereas leukocyte reduced or irradiated and leuko-reduced packed red blood cells were documented in 5% of the PC transfusions. (Table 1).

### 3.2. Diagnosis on Admission and Blood Transfusion

The most common diagnosis for admission was ACS; almost all ACS patients underwent coronary angiography, and the majority also underwent percutaneous coronary intervention (PCI). Nevertheless, there was no increased rate of BT in this subgroup (Table 2). Patients presenting with shock, whether cardiogenic or septic, CHF exacerbation and post trans-catheter aortic implantation (TAVI) were all more likely to receive BT during their admission, as presented in Table 3 and Table 4. Interestingly, patients admitted due to other cardiac interventions, such as mitral clip, electrophysiologic study and implantation of a pacemaker, intracardiac defibrillator (ICD) or cardiac resynchronization therapy (CRT), were less likely to receive BT during their admission.

### 3.3. Length of Stay (LOS) and In-Hospital Complications

Median LOS [IQR] was 5 [2,3,4,5,6] days in the BT-group versus 3 [2,3,4] days in the NBT-group, *p* < 0.001. In-hospital complications were also significantly higher in the BT group and included acute kidney injury (AKI), shock and sepsis, as shown in Table 4.

### 3.4. Mortality Rate

The crude mortality rate was 29.6% in the BT-group and 9.2% in the NBT-group, *p* < 0.0001, as shown in Figure 2. Multivariate Cox analysis for female gender, previous TAVI, cardiac intervention, pacemaker implantation, CHF, sepsis, pulmonary hypertension and low hemoglobin levels found that BT as compared with NBT was independently associated with more than two-fold the mortality rate [HR = 2.19, 95% CI (1.47–3.62), *p* < 0.001] as shown in Table 5. Multivariate analysis ROC curve was plotted and showed an AUC of 0.8, [95%CI (0.760–0.852)] (Figure 3). Multivariate Cox regression model for the number of PCs units received per patient was also independently associated with mortality, with HR of 2.4, 1.5 and 2.5 for 1, 2 and ≥3 PCs units, respectively.

## 4. Discussion

The optimal strategy of BT in ICCU remains controversial, with an increasing number of studies supporting a more restrictive transfusion strategy. Although several studies have tried to address BT in specific situations, such as ACS [13], CHF or post CABG [29], very few have specifically addressed BT in ICCU patients. The present study evaluated the impact of BT on outcomes in a tertiary care center ICCU all-comers population. Our main findings were: (1) Patients treated with BT were older with a higher proportion of females, concurrent comorbidities and in-hospital complications; (2) Patients diagnosed with shock, CHF exacerbation and post TAVI were more likely to receive BT during their admission. Conversely, patients admitted for other cardiac interventions were less likely to receive BT; (3) Even one unit of BT was independently associated with more than two-fold the risk of death; (4) BT was associated with longer length of hospital stay.

Our findings are in accordance with previous studies that found an increased likelihood of BT in elderly patients [26] and females [27]. In a study including patients with non-ST elevation myocardial infarction (NSTEMI) [26], patients of 75 years of age or more had a three-fold increased likelihood of BT than patients <55 years old. Another study evaluating patients with acute coronary syndrome reported that transfused patients were more likely to be elderly and females [27]. Patients undergoing TAVI do not uncommonly require peri-procedural blood transfusion [30]. The multicenter Transfusion Requirements in Transcatheter Aortic Valve Implantation (TRITAVI) registry reported an RBC transfusion rate of 16% in a transfemoral TAVI population [31]. The relative high transfusion rate in our cohort may be attributed to more complex and elderly patients admitted in the ICCU, a sizeable number of procedures performed via alternative vascular access site and thus associated with higher bleeding rates and treating high-risk patients prone to vascular complications.

The independent increased risk of mortality associated with BT has been documented across multiple “real world” cohorts and randomized trials [32,33,34], including ICU [35] and ICCU [36] populations, as in our study. Although considered generally safe and effective, BT can potentially carry considerable risks due to adverse events such as electrolyte disturbances, volume overload and transfusion reactions [37,38,39,40]. However, in high-risk populations, several factors, such as impaired oxygenation, reduced deformability of stored red blood cells, prothrombotic effects due to increased release of procoagulant factors, and transfusion-associated immunosuppression and allosensitization which is known as transfusion related immune modulation(TRIM), have been proposed to explain the association between blood transfusions and poor prognosis [17,34]. Furthermore, BTs are often administered in response to bleeding events, which themselves confer additional risk [26]. Our findings further support the available data, including the evidence of the detrimental effects of BT seen from the very first administered RBC unit.

The potentially harmful effects of BT have led to the adoption of a more cautious transfusion approach, with current guidelines recommending a restrictive BT strategy [i.e., limiting the transfusion of packed red blood cells to hemodynamically stable patients with a Hb value <7 g/dL] in non-bleeding, critically ill patients [22,41]. The restrictive transfusion strategy was found to be non-inferior to a liberal blood administration approach, also in patients with acute MI and Hb level < 8 g/dL [13]. However, proper implementation of recommendations onto specific subsets of high-risk populations is still rather challenging and often requires a personalized approach [42].

Finally, despite all precautionary measures including fluoroscopy- and US-guided vascular punctures, smaller gauge needles used for vascular access, activated clotting time (ACT) monitoring and experienced interventional cardiologists and ICCU intensivists, iatrogenic vascular complications leading to bleeding and anemia are not uncommon. Unfortunately, a recent study reported no benefit of US-guided arterial access used for cardiac interventions in reducing bleeding or vascular complications [43]. Consequently, additional strategies to make vascular access safer are needed.

Limitations: Our study had several limitations; first, we did not stratify patients according to the BT indication. Hence, the mortality rate recorded in our cohort may underestimate the true fatality rate in patients receiving BT following acute bleeding compared with BT administered following other indications. Second, it is a single-center study with its inherent biases; nevertheless, our study consists of a large number of consecutive patients in a real life setting of a tertiary care center ICCU. Third, the causes of death have not been systematically collected. Lastly, we investigated only RBC transfusions. The administration of other blood products has not been studied.

Conclusion: In a contemporary tertiary ICCU delivering guidelines-oriented care, even a single BT remains an independent strong predictor for poor outcomes. Thus, guidelines for further refining the strategy of BT administration in the ICCU setting, including different subsets of high-risk patients, may be warranted.

## Figures and Tables

**Figure 1 jcm-12-01304-f001:**
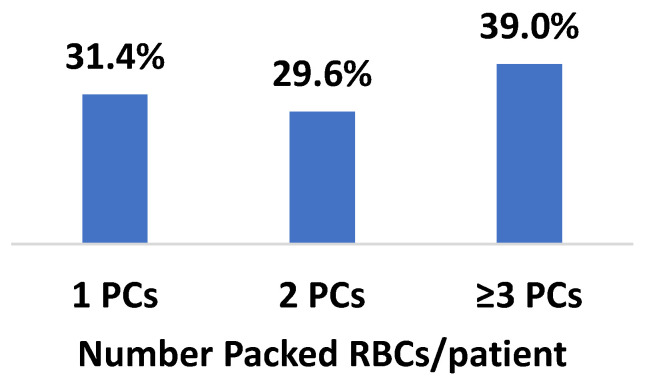
Number of BT units per patient during their admission. RBC = red blood cells.

**Figure 2 jcm-12-01304-f002:**
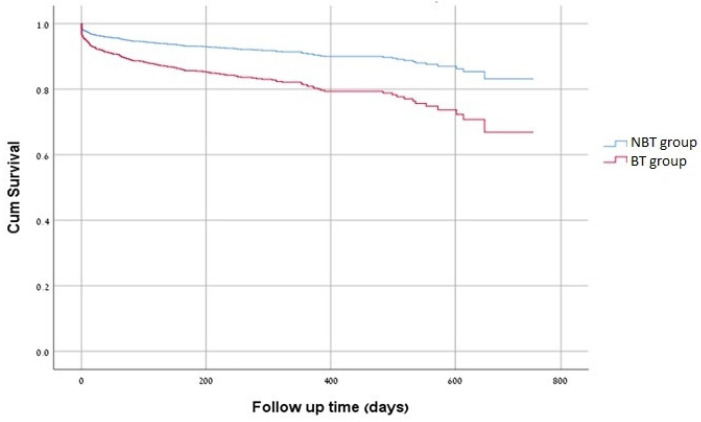
Multivariate Cox Proportional Hazards Analysis. HR = 2.19, 95% CI [1.47–3.62]; *p* < 0.0001.

**Figure 3 jcm-12-01304-f003:**
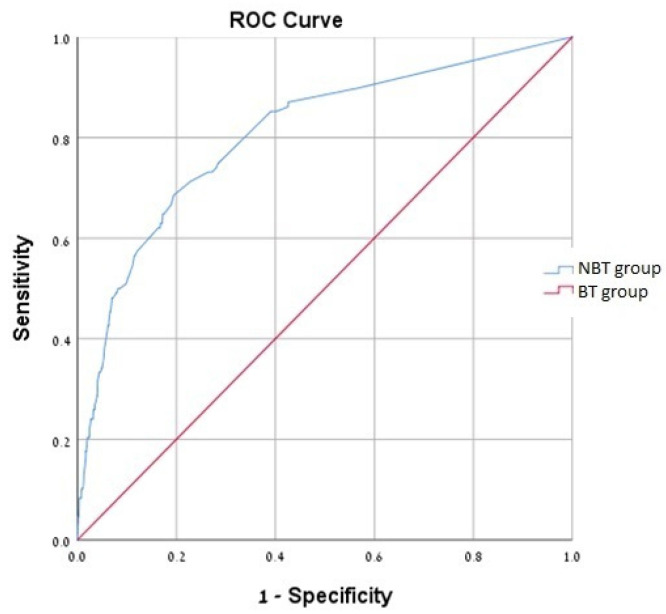
Receiver operating characteristic curve, AUC = 0.806–95% CI [0.760–0.852].

**Table 1 jcm-12-01304-t001:** Patients’ characteristics according to blood transfusion vs. non-blood transfusion.

Clinical Variables	All Patients (*n* = 2132)	Blood Transfusion Group (*n* = 108) (5%)	Non-Blood Transfusion Group (*n* = 2024) (95%)	*p*-Value
Age in years (mean)	66.9 ± 16	73.8 ± 14	66.6 ± 16	<0.0001
Female sex—no. (%)	650 (30.5%)	52 (48.1%)	598 (29.5%)	<0.0001
BMI—Median [IQR]	27.4 [24.3–31.1]	25.9 [22.5–30.9]	27.5 [24.4–31.2]	0.026
Hypertension	1283 (60.2%)	75 (69.4%)	1208 (59.7%)	0.043
DM	757 (35.5%)	47(43.5%)	710 (35.1%)	0.074
Hyperlipidemia	1097 (51.5%)	58 (53.7%)	1039 (51.3%)	0.631
Smoking	597 (28%)	22 (20.4%)	575 (28.4%)	0.07
Prior CAD	602 (28.2%)	40 (37%)	562 (27.8%)	0.037
Prior CABG	132 (6.2%)	13 (12%)	119 (5.9%)	0.01
CVA	150 (7%)	11 (10.2%)	139 (6.9%)	0.189
PAD	90 (4.2%)	9 (8.3%)	81 (4%)	0.029
CHF	300 (14.1%)	38 (35.2%)	262 (12.9%)	0.0001
COPD	168 (7.9%)	17 (15.7%)	151 (7.5%)	0.002
Pulmonary HTN	103 (4.8%)	19 (17.6%)	84 (4.2%)	<0.0001
PE	25 (1.2%)	2 (1.9%)	23 (1.1%)	0.501
Atrial fibrillation	284 (13.3%)	26 (24.1%)	258 (12.7%)	0.001
Anemia	95 (4.5%)	28 (25.9)	67 (3.3%)	<0.0001
CKD	256 (12%)	32 (29.6%)	224 (11.1%)	<0.0001
Cognitive decline	70 (3.3%)	7 (6.5%)	63 (3.1%)	0.056

BMI = body mass index; DM = diabetes mellitus; CAD = coronary artery disease; CABG = coronary artery bypass graft; CVA = cerebrovascular accident; PAD = peripheral artery disease; CHF = congestive heart failure; COPD = chronic obstructive pulmonary disease; PE = pulmonary embolism; CKD = chronic kidney disease; HTN = hypertension.

**Table 2 jcm-12-01304-t002:** Blood transfusion vs. non-blood transfusion according to diagnosis on admission.

Clinical Variables	All Patients (*n* = 2132)	Blood Transfusion Group (*n* = 108) (5%)	Non-Blood Transfusion Group (*n* = 2024) (95%)	*p*-Value
ACS	967 (45.4%)	25 (23.1%)	942 (46.5%)	<0.0001
NSTEMI/UAP	495 (23.2%)	16 (14.8%)	479 (23.7%)	0.033
STEMI	472 (22.1%)	9 (8.3%)	463 (22.9%)	<0.0001
CHF	209 (9.8%)	18 (16.7%)	191 (9.4%)	0.014
Cardiogenic Shock	62 (2.9%)	8 (7.4%)	54 (2.7%)	0.004
Septic Shock	22 (1%)	4 (3.7%)	18 (0.9%)	0.005
VT/VF	57 (2.7%)	5(4.6%)	52 (2.6%)	0.196
AF/AFL	14 (0.7%)	0 (0%)	14 (0.7%)	0.386
Brady-Arrythmia	176 (8.3%)	4 (3.7%)	172 (8.5%)	0.078
Cardiac Tamponade	24 (1.1%)	0 (0%)	24 (1.2%)	0.255

ACS = Acute coronary syndrome; NSTEMI = non-ST-elevation myocardial infarction; UAP = unstable angina pectoris; STEMI = ST elevation myocardial infarction; CHF = congestive heart failure; VT = ventricular tachycardia; VF = ventricular fibrillation; AF = atrial fibrillation; AFL = atrial flutter.

**Table 3 jcm-12-01304-t003:** Blood transfusion rate according to cardiac procedure.

Clinical Variables	All Patients (*n* = 2132)	Blood Transfusion Group (*n* = 108) (5%)	Non-Blood Transfusion Group (*n* = 2024) (95%)	*p*-Value
TAVI	284 (13.3%)	35 (32.4%)	249 (12.3%)	<0.0001
Mitral Clip	27 (1.3%)	2 (1.9%)	25 (1.2%)	0.577
EPS	44 (2.1%)	2 (1.9%)	42 (2.1%)	0.874
Pacemaker/ICD/CRT	5 (0.2%)	0 (0%)	5 (0.2%)	0.605

TAVI = transcutaneous aortic valve intervention; EPS = electrophysiological study; ICD = implantable cardioverter-defibrillator; CRT = cardiac resynchronization therapy.

**Table 4 jcm-12-01304-t004:** Blood transfusion rate according to in-hospital complications.

Clinical Variables	All Patients (*n* = 2132)	Blood Transfusion Group (*n* = 108)	Non-Blood Transfusion Group (*n* = 2024)	*p*-Value
Acute kidney injury	85 (4%)	21 (19.4%)	64 (3.2%)	<0.001
Shock, any	100 (4.7%)	31 (28.7%)	69 (3.4%)	<0.001
Congestive heart failure	76 (3.6%)	10 (9.3%)	66 (3.3%)	<0.001
Sepsis	36 (1.7%)	11 (10.2%)	25 (1.2%)	<0.001
Anoxic brain damage	7 (0.6%)	3 (5.0%)	4 (0.4%)	0.004

**Table 5 jcm-12-01304-t005:** Multivariable regression model for blood transfusion during hospitalization.

Clinical Variable	*p*-Value	Odds Ratio	95% CI	
Female Sex	0.030	1.594	1.045	2.431
Sepsis	0.025	2.861	1.141	7.174
History of CHF	0.014	1.824	1.129	2.946
History of Pulmonary HTN	0.055	1.846	0.987	3.453
History of Pacemaker/ICD	0.024	2.129	1.104	4.106
History of Anemia	<0.001	6.069	3.548	10.378
TAVI During Admission	0.05	1.705	1.000	2.908
Any Intervention	0.003	3.622	1.552	8.452

CHF = congestive heart failure; HTN = hypertension; ICD = Implantable cardioverter-defibrillator; TAVI = transcutaneous aortic valve intervention.

## Data Availability

Data provided upon request.

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
