# Peer review of "Outcomes of Patients Treated with Blood Transfusion in a Contemporary Tertiary Care Medical Center Intensive Cardiac Care Unit"

_jcm, 2023, doi:10.3390/jcm12041304_

Round 1

Reviewer 1 Report

In this study the Authors have analyzed the impact of blood transfusion on outcomes, complications and length of stay of patients admitted to an intensive cardiac care unit. The patients were followed-up for up to 2 years.

Blood transfusion was associated to a greater risk of mortality (dose-depending) and complication as AKI, shock, CHF, sepsis and anoxic brain damage. Moreover, the length of stay of transfused patients was significantly longer.

First of all, I congratulate the Authors for this technically and methodogically sound study. It is of great relevance in the field of interest.

Data are clearly presented and results are robust.

The two-year follow-up represents a further strong point of the study.

I have no main remarks to do.

Page 2 – lines 54 – 56

I suggest to change this sentence as hemolysis and anaphylaxis are only exceptional consequences of BT. The same for transmission of infections as HVB, HIV or HCV.

On the other side, of major concern is Transfusion Associated Cardiac Overload (TACO) , one of the main transfusion-induced cause of mortality.

Among “occult” side effects, immunosuppression is probably the main and storage lesions contribute to this effect.

Author Response

Dear Sir/Madam

We are very thankful for your consideration for publication of our manuscript entitled “Outcomes of patients treated with blood transfusion in a contemporary tertiary care medical center intensive cardiac care unit” at your journal.

I would also to thank the reviewers for the reports and their comments are welcomed.

Regarding report 1:

  • The sentence regarding the side effects was rephrased as suggested by the reviewer.

Reviewer 2 Report

Congratulations for the paper.

references should be updated (there are several "old" ones) it sounds strange not to mention recent TRICS III, for example

Although in the discussion section is mentioned in the introduction talks about blood adverse effect without mentioning the immunomodulatory problem .

Through the paper we are not able to know if in this ICCU they have a liberal or restrictive approach (it is not explained how a BT is ordered)

We don't have any information in the red cell concentrate type (leukorreduces, without Buffy-coat....)

Author Response

Dear Sir/Madam,

We are very thankful for your consideration for publication of our manuscript entitled “Outcomes of patients treated with blood transfusion in a contemporary tertiary care medical center intensive cardiac care unit” at your journal.

I would also to thank the reviewers for the reports and their comments are welcomed.

Regarding report 2:

  • References were updated as requested, with an emphasis on recent trials including the TRICS III.
  • The immunomodulatory effect related to transfusion was also mentioned in the introduction section as requested.
  • Regarding the approach that was adopted for blood transfusion at out center, we rephrased it in the method section as a restrictive approach.
  • We also provided information about the red cell concentrate that we gave to our patients.

Thank you very much

Kind regards,

Hani Karameh, MD. MRCP

Jesselson Integrated Heart Center, Shaare Zedek Medical Center
